# Improving Health Literacy: Analysis of the Relationship between Residents’ Usage of Information Channels and Health Literacy in Shanghai, China

**DOI:** 10.3390/ijerph19106324

**Published:** 2022-05-23

**Authors:** Ya Gao, Chen Chen, Hong Hui, Mingyue Chen, Ning Chen, Hong Chen, Weiming Zeng, Yan Wei, Zhaoxin Wang, Jianwei Shi

**Affiliations:** 1School of Public Health, Shanghai Jiaotong University School of Medicine, Shanghai 200025, China; gaoya1@sjtu.edu.cn (Y.G.); vita_cmy@sjtu.edu.cn (M.C.); cheneycnn@163.com (N.C.); chenhong199706@163.com (H.C.); wmzeng0@sjtu.edu.cn (W.Z.); 2Shanghai Jing’an District Jiangning Road Community Health Service Centre, Shanghai 200041, China; cc13916627775@163.com; 3Shanghai Bao Shan District Gucun Town Community Health Service Centre, Shanghai 201906, China; huihonghyd1@163.com; 4Key Lab of Health Technology Assessment (National Health Commission), School of Public Health, Fudan University, Shanghai 200032, China; yanwei@fudan.edu.cn

**Keywords:** health literacy, information channels, health communication, China

## Abstract

Background: This study aimed to examine the relationship between residents’ health literacy (HL) and their use of and trust in information channels. Methods: A community-based cross-sectional health survey utilizing a cluster sampling design was conducted in January 2022. The sample consisted of 1067 residents in Shanghai, China. Those who correctly answered over 80% of the questions were regarded as qualified. The differences in residents’ HL and the dimensions of knowledge HL, lifestyle HL, and skills HL were analyzed based on their use of and trust in traditional media, the internet, and offline activities. Logistic regression was conducted to examine the effects of the usage of these channels on all four types of HL. Results: A total of 27.65% of participants were qualified for HL. The use of traditional media (OR = 1.405, *p* < 0.05) and engagement in offline activities (OR = 1.951, *p* < 0.05) were significantly related to HL. Disbelief in traditional media was related to being qualified in knowledge HL (OR = 1.262; *p* < 0.05), whereas disbelief in offline activities had an adverse effect on knowledge HL and skills HL (OR = 0.700, 0.807; *p* < 0.05). Conclusion: Effort should be made to improve the efficiency of offline health education, and ensure the reliability and quality of health-related information from mass media and the internet to improve residents’ HL.

## 1. Introduction

According to the WHO, health literacy (HL) represents an individual’s cognitive and social skills to access, understand, and use information to promote and maintain well-being [1]. Existing studies have shown that HL is an important determinant of population health. For instance, a study by the American Medical Association showed that HL was highly associated with multiple aspects of health, including health knowledge, health status, and the use of health services [2]. Health literacy is also correlated with quality of life in the dimensions of physical and mental health [3]. However, limited HL is consistently a global health problem. In the European Health Literacy Survey (HLS-EU), which was conducted in eight European Union countries, approximately one half of the participants had limited (insufficient or problematic) HL [4]. In China, in 2020, statistics showed that 23.15% of residents aged 15–69 were qualified in HL, which meant that these populations met the requirement for basic HL. These results indicate that improving HL should be emphasized in these countries.

Recent studies have revealed that HL is associated with health communication and health information. In Nutbeam’s conceptual model of health literacy as an asset, HL is the outcome of health education and communication [5]. Tailored information, communication, and education help to develop knowledge and capability, leading to the improvement of HL. According to the simplified HL model by Gillian Rowlands, in addition to family history and ethnicity/culture, the key factors of HL are “collecting health information” and “the way you live your life”, with “collecting health information” being modulated by “the health information environment”, including information from health services, friends and family, libraries, and the media [6]. Cross-sectional surveys and a model with acceptable fit showed that access to health information is one of the predictive factors of HL [7,8,9]. However, only a limited number of these studies asked the participants about their use of each channel of health information and their trust in information channels [8]; therefore, whether the use of and trust in common information channels affect the level of HL is unknown. Although the impact of health information use on the level of HL has been discussed in other countries, there is a lack of evidence in Chinese residents. Only a limited number of studies have been conducted with certain groups of Chinese people regarding HL status, and the relationship between the usage of information channels and HL among Chinese residents has not been well described [10,11,12].

The purpose of this study was to explore the relationship between the use of and the degree of trust in health information channels and the qualification in HL knowledge, lifestyle, and skills in Shanghai, China. This study provides a reference and guidance for health communication workers to employ targeted publicity tools according to the usage habits of residents.

## 2. Materials and Methods

### 2.1. Participants and Data Collection

This study, conducted in January 2022, was a community-based cross-sectional health survey utilizing a cluster sampling design in Shanghai, China. We enrolled residents of Jing’an District as the study population, and selected three communities, each of which included over 15 resident committees in the district. We randomly coded and selected 1 residential building from each resident committee, and all the residents in these buildings were included in the investigation. The criteria for the survey subjects were as follows: (1) they were part of the permanent population of Shanghai, that is, they had lived in Shanghai for at least half a year; and (2) they volunteered to participate in the investigation. Participants were excluded from the survey if they were on a business trip or unable to finish the online questionnaire. To be consistent with the Chinese Residents Health Literacy Monitoring Program, which covers people aged 15–69, we found it necessary to ensure the number of teenage participants. We randomly selected three schools in the area, contacted the person in charge, and distributed questionnaires to the students. We collaborated with the community health centers (CHCs) in the area responsible for chronic disease management and primary health care and the schools to collect the data.

### 2.2. Study Questionnaires and Additional Questions

The questionnaire of the cross-sectional survey consisted of 14 questions referring to health communication, and 44 questions evaluating HL in addition to sociodemographic factors. The sample size needed to statistically cover 10 times the number of items, with a minimum sample size of 580. In this study, we set the sample size as 20 times 1160, considering various groups of residents. Furthermore, considering a nonresponse rate of 10%, the actual sample size increased to 1160/0.9 = 1288.89, rounded to 1289. The finished questionnaires were collected by the researcher; the return of the questionnaire implied consent to participate. A total of 1289 questionnaires were issued, and the valid response rate was 82.8%.

The questionnaire consisted of three parts: (1) sociodemographic variables, including age, gender, education level, monthly income, marital status, and health condition, including a health self-assessment and number of chronic diseases; (2) the use of and degree of trust in different information channels; and (3) the evaluation of HL. Multiple-choice questions were used in the first and second parts. Seven information channels were listed in the second part, including TV and radio, books and newspapers, apps and social networks, healthcare applications and online communities, relatives and friends, community publicity and knowledge lectures, and doctors and other professionals. Participants were asked to select the channels they used. These seven channels were combined into three categories in the data analysis: (1) traditional media, including TV and radio, and books and newspapers; (2) the internet, including apps and social networks, and healthcare applications and online communities; and (3) offline activities, including relatives and friends, community publicity and knowledge lectures, and doctors and other professionals. Thus, the questionnaire involved three questions to evaluate residents’ trust in the three categories, each of which used a Likert scale ranging from 1–5 points, corresponding to total belief, basic belief, partial belief, basic disbelief, and total disbelief. The third part of the questionnaire used in the study was designed based on the Chinese resident HL monitoring questionnaire by the Chinese Health Education Centre, which is updated every year. According to the Chinese Health Education Centre, HL includes three dimensions: knowledge, lifestyle, and skills. We selected 44 overlapping questions from the 2012 questionnaire and the 2020 questionnaire to reduce the number of questions and facilitate the evaluation of the three dimensions of HL [13,14].

The score for each single-choice and true-false question was 1 point, and that for each multiple-choice question was 2 points. The respondents who correctly answered over 80% of all the questions were regarded as qualified in HL, and those who correctly answered over 80% of the questions in a certain dimension were regarded as qualified in that dimension. The total scores of HL and the three dimensions were 56, 25, 22, and 9, respectively, and the passing scores were 45, 20, 18, and 7, respectively. The questionnaire had a Cronbach’s α coefficient of 0.731 and a Guttman split-half coefficient of 0.723, indicating good reliability and good content validity.

### 2.3. Statistical Analysis

SPSS software (SPSS 25.0, Chicago, IL, USA) was used for data analysis. A chi-squared test was used to compare HL among groups with different sociodemographic characteristics and different usages of information channels.

To facilitate the analysis, trust in the three information channels was taken as a continuous variable, and the Mann–Whitney U test was used, since the trust of the participants was not normally distributed. Logistic regression was used to analyze the possible influencing factors of HL and the three HL dimensions. A *p* < 0.05 was considered to be statistically significant.

## 3. Results

### 3.1. Descriptive Statistical Analysis

A total of 1067 valid questionnaires were collected and analyzed. The general characteristics of the study population are presented in Table 1. Among the 1067 participants, 39.36% were male and 60.64% were female. The participants were divided into four groups based on their ages: <18, 18–39, 40–59, and ≥60. The majority of participants were married (62.60%), had a high level of education (50.80%), had good health self-assessments (57.36%), and had no chronic diseases (75.54%). In total, 27.65% of the participants were qualified in HL, and 37.86%, 27.84%, and 37.39% of the participants were qualified in knowledge, lifestyle, and skills HL, respectively (shown in Figure 1). More people were qualified in knowledge HL, and fewer people were qualified in lifestyle HL. Significant differences were found in age, gender, education level, per capita income, and marriage status between the qualified and unqualified HL groups (*p* < 0.05). The number of people who were qualified in HL and knowledge HL increased significantly in participants aged 18–39 and 40–59, whereas the number of people who were qualified in skills HL increased significantly in those over 40. Among females and those with a high educational level and high per capita income, more people were qualified for HL. Unmarried participants had lower scores on the survey.

We compare the use of and trust in three categories of information channels between qualified and unqualified groups in Table 2. For all four types of HL, qualified participants accounted for a greater proportion of internet users (67.1%, 66.6%, 70.4%, 64.7%; *p* < 0.05) and offline activities (67.1%, 61.6%, 64.3%, 63.4%; *p* < 0.05). No significant differences were observed in traditional media. Higher trust in offline activities was related to a higher possibility of being qualified in knowledge HL and skills HL (*p* < 0.05).

### 3.2. Association between Health Literacy and Information Channels

The results of multiple multivariate logistic regressions of the association between the use of and trust in information channels and HL are shown in Table 3. Each regression controlled for the participants’ age, gender, education, income, marital status, health self-assessment, and number of chronic diseases. Participants who received health information from traditional media were more likely to be qualified in HL (OR = 1.405; 95% CI = 1.003–1.970, *p* < 0.05), and the dimensions of knowledge (OR = 1.403; 95% CI = 1.024–1.924, *p* < 0.05) and skills (OR = 1.491; 95% CI = 1.099–2.022, *p* < 0.05); however, disbelief in traditional media was related to being qualified in knowledge HL (OR = 1.262; 95% CI = 1.016–1.567, *p* < 0.05). Those who had engaged in offline activities were more likely to be qualified in all four types of HL (OR = 1.951; 95% CI = 1.432–2.656; OR = 1.514; 95% CI = 1.142–2.007; OR = 1.638; 95% CI = 1.212–2.213; OR = 1.775; 95% CI = 1.350–2.334, *p* < 0.05), and disbelief had an adverse effect on knowledge HL (OR = 0.070; 95% CI = 0.569–0.860, *p* < 0.05) and skills HL (OR = 0.807; 95% CI = 0.661–0.985, *p* < 0.05). Users of the internet were more likely to be qualified in lifestyle HL (OR = 1.582; 95% CI = 1.157–2.161, *p* < 0.05) and skills HL (OR = 1.342; 95% CI = 1.013–1.778, *p* < 0.05).

## 4. Discussion

### 4.1. Levels of Health Literacy

In our study, 27.65% of participants were qualified in HL. The proportions of participants who were qualified in knowledge HL, lifestyle HL, and skills HL were 37.86%, 27.84%, and 37.39%, respectively, which were higher than a previous study and the national statistics for 2020 [15,16]. One possible explanation is that the questionnaires used in each study had slight differences, as the purposes were different. Generally, it has been suggested that the majority of residents in China do not meet the requirement for basic HL, whereas residents in developed countries show higher HL levels [17,18,19]. The proportion of participants with limited HL in the European Health Literacy Survey (HLS-EU) was 47%, and the proportion was 13% in a national cross-sectional community survey in Australia [4,20]. This comparison highlights the low HL in China. Considering economic factors and educational levels, residents’ HL might be even lower in many other regions. Improving HL presents a challenge to researchers, community organizations, health care providers, and policy-makers.

### 4.2. Socioeconomic Determinants of Health Literacy

Education level had a statistically significant effect on general HL and the three dimensions of HL. Higher HL was associated with a higher education level in all models. This result is consistent with previous studies showing that educational level is a determinant of HL [9,21,22,23]. Other studies conducted worldwide have demonstrated that HL is higher in people with higher perceived financial status or household income [9,22,24,25,26]. The significant HL differences among the education and income groups show that HL reflects social inequalities.

The regression models show that females have an advantage in terms of HL knowledge. A previous study in Shanghai showed no association between gender and HL [15], and other studies obtained different results [9,15,19,21,25,26]. A possible reason is that in a social sense, gender is constructed based on different societal conditions [23]. The general characteristics may still be confounding factors, and further studies may help to reveal the specific mechanisms.

### 4.3. Information Channels and Health Literacy

In our study, the use of traditional media was related to HL, and the knowledge and skills HL dimensions. Receiving health-related information from traditional media is common in China: 730 participants in our study used traditional media, including TV and newspapers. A nationwide survey in Turkey also found that although the internet had replaced traditional media as the most commonly used information channel, the use of traditional media such as newspapers and TV still contributed to a higher HL score [23]. In our study, there was no significant correlation between the use of traditional media and a healthy lifestyle. In Spain, a randomized intervention study of the HAVISA plan also found that health messages in television food advertisements could not change the attitudes or immediate eating behaviors of adolescents [27]. A possible reason is that traditional media offer one-way health communication to the population, and lifestyle-related information in mass media may cater more to the mass market. One example is the healthy habits of eating light food and obtaining adequate sleep, which are applicable to people of all ages and health statuses. However, with increasing economic and work pressure, people’s desire for health becomes personalized, and people tend to actively search for and adopt suggestions that fit their daily lives [28,29,30]. Regarding levels of trust, participants who did not believe in traditional media were more likely to be qualified in knowledge HL, which was not discussed by previous studies. Mass media contains information that offers health guidance, as well as implicit and explicit relevant content from commercial entities or health systems that can be either health-promoting or health-compromising [31]. Therefore, researchers have proposed the concept of media health literacy, and noted that it is an important determinant of health literacy [31,32,33,34]. A lack of originality and excessive advertisements or entertainment factors used to be serious limitations of China’s healthcare television programs, which affected the authenticity of health knowledge and reduced public reliability. In 2014, the State Administration of Press, Publication, Radio, Film, and Television issued a notice on the production and broadcasting of healthcare TV programs, which stopped entrepreneurs from producing these programs, and established the requirements for the professionalism of guests and hosts to standardize China’s healthcare TV programs and enhance the public’s trust in them [35]. To facilitate health communication, researchers and workers in this field should pay more attention to tracking the trends of people’s health needs in a timely manner, improve the practicability of information to maintain credibility, and eliminate misleading advertisements in mass media.

Surprisingly, no significant correlation was found between the use of the internet and HL in this study, although the results from other studies showed that people with adequate HL were more likely to seek online health information [23,36,37]. The degree of trust in the internet also had no significant effect on HL. However, another study suggested that trust in doctors’ social media was one of the specific mediators between health literacy and preventive behaviors during the COVID-19 pandemic [38]. Information about health and diseases is widely available on the internet with the increasing use of mobile devices, which facilitate access to the internet and health-related websites. However, despite the current emphasis on standardizing health communication and information supervision on the internet, there is still a large amount of unproven, low-quality information (including health knowledge and lifestyle suggestions) on social media and apps that does not promote HL [28]. There is also hidden danger of information leaks on the internet, since users of medical websites need to register with personal information, and online communication with doctors is not part of a patient’s formal medical record [28,39]. Such concerns may cause distrust in the internet and hinder internet use. Therefore, to improve the HL of the entire population, the government and health workers need to consider how to guide people to obtain high-quality health information from credible channels, and how to prevent the impact of security vulnerabilities.

Our study demonstrated a strong association between HL and offline activities, which has not been reported previously. Offline activities are relatively reliable in China, since the guests who are invited by communities to hold free clinics and give lectures are generally clinicians or experts. It might be difficult for those who do not believe in offline activities to become qualified in knowledge HL and skills HL. A cross-sectional survey in the United States also showed that people with low HL were less likely to trust health information from specialist doctors [40]. These people may experience communication problems and/or may not have previously received full healthcare information, resulting in a negative impression of doctors and other professionals [41]. In other countries, activities and intervention programs are important means of health education. For instance, the United States has promoted the health education program Whole School, Whole Community, and Whole Child nationwide since 2014, and has maintained a high degree of collaboration among schools, families, and communities to effectively promote the effect of health education [42]. However, few communities in Shanghai have conducted such activities for all residents, except health management for patients with chronic diseases, and women and children. In addition, people with adequate HL are generally younger, which means they work full-time and may not have spare time for such activities. Neither the number of health education activities nor the number of participants is sufficient. Considering all these factors, it is essential for communities to improve the means and efficiency of health education. For instance, experts can be invited to hold online lectures and meetings to enhance residents’ participation. Inviting experts to broadcast online or record relevant educational lectures is also an innovative way to enhance residents’ participation.

### 4.4. Study Limitations

There are some potential limitations to our study. This study was conducted in a community setting, which may allow its findings to be generalized to the source population. However, as this was a cross-sectional analysis of data, individuals may have faced recall or social desirability bias since they were asked about past events.

## 5. Conclusions

The use of traditional media and the internet and engagement in offline activities were significantly related to HL, and trust in traditional media and offline activities were also significantly related to HL. More efforts should be made to improve the efficiency of offline health education, and ensure the reliability and quality of health-related information from mass media and the internet to improve residents’ HL.

## Figures and Tables

**Figure 1 ijerph-19-06324-f001:**
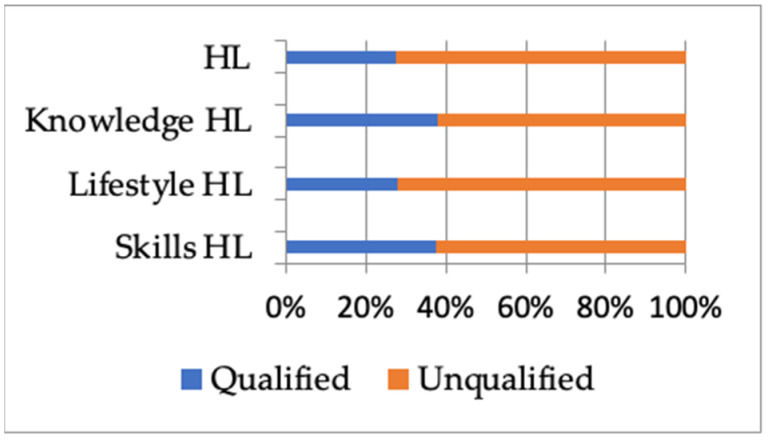
Levels of health literacy and the three dimensions of knowledge, lifestyle, and skills health literacy. (HL = health literacy).

**Table 1 ijerph-19-06324-t001:** General characteristics of participants qualified in health literacy, knowledge HL, lifestyle HL, and skills HL.

Characteristics	Total (*n* = 1067)	HL Qualified Number(%) ^a^	*p* Value	Knowledge HL Qualified Number(%) ^a^	*p* Value	Lifestyle HL Qualified Number(%) ^a^	*p* Value	Skills HL Qualified Number(%) ^a^	*p* Value
Age group			<0.001		<0.001		0.056		<0.001
<18	211	35 (16.6)		60 (28.4)		47 (22.3)		62 (29.4)	
18–39	393	111 (28.2)		162 (41.2)		109 (27.7)		128 (32.6)	
40–59	352	122 (34.7)		152 (43.2)		114 (32.4)		160 (45.5)	
≥60	111	27 (24.3)		30 (27.0)		27 (24.3)		49 (44.1)	
Gender			0.011		0.010		0.037		0.193
Male	420	98 (23.3)		139 (33.1)		102 (24.3)		147 (35.0)	
Female	647	197 (30.4)		265 (41.0)		195 (30.1)		252 (38.9)	
Education level			<0.001		<0.001		<0.001		<0.001
Junior high school or below	182	17 (9.3)		25 (13.7)		17 (9.3)		46 (25.3)	
Senior high school and technical	343	64 (18.7)		96 (28.0)		76 (22.2)		114 (33.2)	
College	167	43 (25.7)		65 (38.9)		42 (25.1)		61 (36.5)	
Graduate and above	375	171 (45.6)		218 (58.1)		162 (43.2)		178 (47.5)	
Per capita monthly income ^b^			<0.001		<0.001		<0.001		0.001
<¥4000	210	35 (16.7)		47 (22.4)		39 (18.6)		53 (25.2)	
¥4000–¥6000	242	63 (26.0)		88 (36.4)		55 (22.7)		99 (40.9)	
¥6000–¥8000	193	53 (27.5)		78 (40.4)		60 (31.1)		72 (37.3)	
≥¥8000	422	144 (34.1)		191 (45.3)		143 (33.9)		175 (41.5)	
Marital status			0.010		0.017		0.164		0.001
Unmarried	357	78 (21.8)		114 (31.9)		90 (25.2)		108 (30.3)	
Married	668	203 (30.4)		274 (41.0)		191 (28.6)		270 (40.4)	
Separated, divorced, or widowed	42	14 (33.3)		16 (38.1)		16 (38.1)		21 (50.0)	
Health self-assessment			0.099		0.798		0.136		0.515
Good	612	156 (25.5)		227 (37.1)		156 (25.5)		221 (36.1)	
Not bad	401	126 (31.4)		155 (38.7)		125 (31.2)		155 (38.7)	
Bad	54	13 (24.1)		22 (40.7)		16 (29.6)		23 (42.6)	
Number of chronic diseases			0.401		0.067		0.562		0.877
0	806	224 (27.8)		321 (39.8)		229 (28.4)		298 (37.0)	
1	198	58 (29.3)		63 (31.8)		54 (27.3)		77 (38.9)	
≥2	63	13 (20.6)		20 (31.7)		14 (22.2)		24 (38.1)	

^a^ percentage of all qualified participants; ^b^ ¥: RMB, Chinese Yuan.

**Table 2 ijerph-19-06324-t002:** Distribution and comparison of use of and trust in information channels according to health literacy type.

Characteristics	Total (*n* = 1067)	HL Qualified Number(%) ^a^	*p* Value	Knowledge HL Qualified Number(%) ^a^	*p* Value	Lifestyle HL Qualified Number(%) ^a^	*p* Value	Skills HL Qualified Number(%) ^a^	*p* Value
Use of traditional media			0.788		0.644		0.228		0.173
No	337	95 (32.2)		131 (32.4)		102 (34.3)		116 (29.1)	
Yes	730	200 (67.8)		273 (67.6)		195 (65.7)		283 (70.9)	
Levels of trust			0.514		0.059		0.901		0.77
Total trust to total disbelief (1–5) ^b^		2.46 ± 0.713		2.49 ± 0.734		2.45 ± 0.734		2.42 ± 0.708	
Use of internet			<0.001		<0.001		<0.001		0.002
No	442	97 (32.9)		135 (33.4)		88 (29.6)		141 (35.3)	
Yes	625	198 (67.1)		269 (66.6)		209 (70.4)		258 (64.7)	
Level of trust			0.553		0.666		0.737		0.822
Total trust to total disbelief (1–5) ^b^		2.47 ± 0.627		2.46 ± 0.673		2.46 ± 0.667		2.45 ± 0.659	
Engagement in offline activities			<0.001		<0.001		<0.001		<0.001
No	516	97 (32.9)		155 (38.4)		106 (35.7)		146 (36.6)	
Yes	551	198 (67.1)		249 (61.6)		191 (64.3)		253 (63.4)	
Level of trust			0.11		0.018		0.192		0.026
Total trust to total disbelief (1–5) ^b^		2.16 ± 0.659		2.15 ± 0.667		2.17 ± 0.682		2.15 ± 0.642	

^a^ percentage of all qualified participants; ^b^ 1–5 points corresponding to total belief, basic belief, partial belief, basic disbelief, and total disbelief, respectively.

**Table 3 ijerph-19-06324-t003:** Logistic analysis of the predictors of health literacy.

Characteristics	HL OR (95% CI)	*p* Value	Knowledge HL OR (95% CI)	*p* Value	Lifestyle HL OR (95% CI)	*p* Value	Skills HL OR (95% CI)	*p* Value
Age group		0.037		0.298		0.069		0.004
<18	1		1		1		1	
18–39	1.122 (0.561, 2.244)		0.751 (0.405, 1.393)		0.835 (0.430, 1.623)		0.995 (0.546, 1.814)	
40–59	1.958 (0.931, 4.118)		1.007 (0.513, 1.977)		1.394 (0.679, 2.864)		1.883 (0.975, 3.635)	
≥60	1.775 (0.760, 4.145)		0.74 (0.338, 1.624)		1.438 (0.630, 3.278)		2.039 (0.967, 4.298)	
Gender		0.057		0.044		0.132		0.708
Male	1		1		1		1	
Female	1.351 (0.991, 1.842)		1.339 (1.008, 1.779)		1.261 (0.932, 1.706)		1.054 (0.801, 1.386)	
Education level		<0.001		<0.001		<0.001		0.001
Junior high school or below	1		1		1		1	
Senior high school and technical	2.283 (1.191, 4.373)		2 (1.140, 3.511)		2.446 (1.284, 4.662)		1.489 (0.913, 2.428)	
College	3.327 (1.722, 6.429)		3.704 (2.095, 6.550)		3.278 (1.692, 6.351)		1.634 (0.976, 2.733)	
Graduate and above	7.685 (4.111, 14.363)		7.482 (4.335, 12.915)		6.842 (3.659, 12.793)		2.553 (1.573, 4.143)	
Per capita monthly income ^a^		0.605		0.265		0.559		0.074
<¥4000	1		1		1		1	
¥4000–¥6000	1.236 (0.741, 2.059)		1.536 (0.967, 2.439)		0.854 (0.515, 1.417)		1.756 (1.137, 2.714)	
¥6000–¥8000	1.086 (0.630, 1.872)		1.408 (0.859, 2.308)		1.136 (0.675, 1.910)		1.419 (0.882, 2.284)	
≥¥8000	1.328 (0.821, 2.147)		1.501 (0.968, 2.328)		1.12 (0.704, 1.784)		1.58 (1.035, 2.414)	
Marital status		0.604		0.426		0.142		0.246
Unmarried	1		1		1		1	
Married	0.857 (0.536, 1.371)		1.32 (0.850, 2.051)		0.799 (0.503, 1.271)		1.078 (0.694, 1.675)	
Separated, divorced, or widowed	1.175 (0.506, 2.730)		1.479 (0.655, 3.340)		1.557 (0.690, 3.512)		1.869 (0.867, 4.028)	
Health self-assessment		0.608		0.454		0.319		0.495
Good	1		1		1		1	
Not bad	1.175 (0.856, 1.615)		0.984 (0.731, 1.326)		1.217 (0.891, 1.661)		0.971 (0.728, 1.294)	
Bad	1.078 (0.512, 2.270)		1.515 (0.769, 2.985)		1.494 (0.744, 3.002)		1.426 (0.755, 2.695)	
Number of chronic diseases		0.42		0.308		0.612		0.49
0	1		1		1		1	
1	1.049 (0.691, 1.594)		0.74 (0.494, 1.107)		0.977 (0.644, 1.482)		0.902 (0.616, 1.319)	
≥2	0.627 (0.294, 1.338)		0.758 (0.381, 1.508)		0.69 (0.328, 1.448)		0.68 (0.357, 1.295)	
Use of traditional media		0.048		0.035		0.656		0.01
No	1		1		1		1	
Yes	1.405 (1.003, 1.970)		1.403 (1.024, 1.924)		1.077 (0.777, 1.492)		1.491 (1.099, 2.022)	
Level of trust		0.643		0.035		0.959		0.955
Total trust to total disbelief (1–5)	1.057 (0.835, 1.339)		1.262 (1.016, 1.567)		0.994 (0.787, 1.256)		1.006 (0.814, 1.243)	
Use of internet		0.155		0.104		0.004		0.04
No	1		1		1		1	
Yes	1.256 (0.917, 1.720)		1.272 (0.952, 1.701)		1.582 (1.157,2.161)		1.342 (1.013, 1.778)	
Level of trust		0.437		0.51		0.562		0.288
Total trust to total disbelief (1–5)	1.098 (0.867, 1.391)		1.075 (0.866, 1.335)		1.071 (0.849, 1.352)		1.121 (0.908, 1.383)	
Engagement in offline activities		<0.001		0.004		0.001		<0.001
No	1		1		1		1	
Yes	1.951 (1.432, 2.656)		1.514 (1.142, 2.007)		1.638 (1.212, 2.213)		1.775 (1.350, 2.334)	
Level of trust		0.097		0.001		0.169		0.035
Total trust to total disbelief (1–5)	0.828 (0.663, 1.035)		0.7 (0.569, 0.860)		0.858 (0.689, 1.067)		0.807 (0.661, 0.985)	

^a^ ¥: RMB, Chinese yuan.

## Data Availability

Data can be obtained from https://doi.org/10.6084/m9.figshare.19793098.v2.

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
