# Peer review of "Improving Health Literacy: Analysis of the Relationship between Residents’ Usage of Information Channels and Health Literacy in Shanghai, China"

_ijerph, 2022, doi:10.3390/ijerph19106324_

Round 1

Reviewer 1 Report

Title

  • The authors should include the study population in the title

Abstract

  • Again, the study population should be included in the method section. Who were the respondents?

Background

  • Relevant information about HL has been provided. However, the information about the study population is lacking. Why did the authors conduct the study in this population? Why was it important?

Methods

  • Why did this study have to collect the data from 2 populations (residents and students)? It was uncommon to do so if we did not aim to compare these two populations. I think it is not appropriate to include all information from both groups together in the analysis. Please consider this issue. This is a major concern. However, if the authors decide not to reanalyze, the background should convince readers that it is appropriate to combine the results from the two populations together.

Results

  • If the analysis has been changed according to the suggestion in the method section, please rewrite this section.

Discussion

  • The overall writing in this section is relevant to the results.
  • I think that one reason why the proportion of qualified HL is different from the other study could be the questionnaires that have been used in each study were different.

Author Response

We greatly appreciate your positive and insightful review to help us improve the manuscript. After careful consideration of your comments and suggestions, we have revised the manuscript in several sections. The changes are highlighted in yellow. Our point-by-point responses are provided below. We hope that the revisions render the manuscript more acceptable. Please see the attachment.Thank you again for your generous contribution.

Reviewer 2 Report

I applaud the authors for conducting an interesting article.  The study does have weaknesses in method section, and the manuscript could be helped with significantly better editing. It also requires grammatical revised.

This manuscript could be an important reference for future studies. However, is still needed to improve the quality of this paper. Please revise the manuscript to address the expressed concerns. After thorough review, I am recommending Major revisions. In this regard, kindly address the following comments and suggestions to further improve your manuscript

  1. Please write the type of study, sample size, sampling strategy and date and country of study in both abstract and method section

  1. It was better if you wrote some of main finding as quantitative or mean ±SD within the abstract. The result section in the abstract is poor and immature!!

  1. The conclusion of the abstract needs to answer the research question and match the study aims

  1. The Introduction is so weak. You could summarize this section a bit more for readers. Write about the problems, the novelty of your study, and your study goals within the introduction. In this section, you can use the following articles:
  • Mental health literacy and quality of life in Iran: a cross-sectional study
  • Designing and Implementing Virtual Education Course of Media Literacy for Medical Sciences Students: An Experimental Study

  1. The novelty of your work must be discussed , it seems it is so weak

  1. The materials & methods section is relatively immature. You could expand it a bit more clearly for readers. For example, write about the sample size. How did you calculate sample size for this survey? Where have you collected samples? Write the year and the name of place in which you had done this survey. Furthermore, write about all applied exclusion criteria a bit more clearly by which you selected samples for this survey.

  1.  What was your sample size formula? What is your expected power? please mention in main text

  1. Discuss more about your sampling strategy? The structure of your sampling is so vague and understandable. Did you have sampling frame? how did you access to this frame

  1. What are the data extract’s center characteristics? is it governmental or private, is it referral or not referral and so on, discuss more about it

  1. You could increase the number of more recently studies in the reference section. You should have comprehensive and reliable comparisons between your findings with the other previous studies. In the discussion section I would like to see a more profound discussion about the findings. What is the meaning of your results in light of earlier studies / theories. The discussion should be more than only a repetition of the results accompanied by some arbitrary studies

 There are some spelling and grammatical errors in the text. Please correct them

Reviewer 3 Report

The paper provides a study exploring the relationship between the use and the degree of trust in health information channels and the qualification in knowledge, lifestyle and skills HL (health literacy) of Shanghai residents and students. In the following my suggestions.

Introduction

As described by the authors in the introduction there is a lack of research about the impact of information channels on the level of HL (p. 2, l. 55-58), however, the references are missing. Furthermore, the statement “(…) and therefore could not elaborate the impact of information channels on the level of HL”. (p. 2, l. 57-58) has to be justified.

Please change (p.2, l. 59-60): “The purpose of the study was to determine whether the use and degree of trust in specific information channels is related to the level of HL”. And, please explain according to theoretical approaches the importance of health communication and health information for the improvement of HL because this is in the focus of the study.   

Please use “trust in different information channels” consequently throughout the paper because “attitude towards” is not the same theoretical approach.

Method

In the chapter 2.1 (“Participants and Data Collection”) the authors did not justify why they selected residents on the one hand and students on the other hand. In the result part they divided the participants in four age groups to study age-specific qualification in HL. This information should begiven in the Method chapter.    

The establishment of the seven information channels is unclear (e.g. the trust in apps in different from the trust in social networks) and also the categorization in three parts is not justified.

The description of the questionnaire about the use of and degree of trust in information channels is too little. This also applies to the description of the questionnaire about the level of HL.

Results

All the tables are confusing and the font size is much too small.

Discussion

References are missing to the statement “(…) residents in developed countries showed higher HL levels” (p. 8, l. 184-185).

“Elderly individuals may not realize that a healthy lifestyle is helpful for maintaining health and preventing diseases (…)” (p. 9, l. 221-223). This argument is not comprehensible, since older people in particular are far more concerned with the topic of health and health behavior than younger people.

Conclusions

“More efforts should be made to encourage the correct use of information channels (…)” (p.10, l. 278-280). It is necessary to highlight which channels are to be addressed here and, above all, in what form they are to contribute to the improvement of HL. This would be a very important aspect to discuss with regard to the knowledge gain of the study.

Round 2

Reviewer 1 Report

-

Reviewer 2 Report

Accept

Reviewer 3 Report

Response to the authors:

The paper has gained considerably in quality due to the additions, especially the added references. 
The methodological procedure within the study is now much more comprehensible for the reader.